# Narciclasine as a Novel Treatment for Lung Cancer and Malignant Pleural Mesothelioma: Insights from 3D Tumor Spheroid Models

**DOI:** 10.3390/ijms262010127

**Published:** 2025-10-17

**Authors:** Sicong Jiang, Serkan Berkcan, Nadja Perriraz-Mayer, Frédéric Triponez, Véronique Serre-Beinier

**Affiliations:** Division of Thoracic and Endocrine Surgery, Department of Surgery, Faculty of Medicine, Geneva University Hospitals, 1205 Geneva, Switzerlandnadja.mayer@unige.ch (N.P.-M.); frederic.triponez@hug.ch (F.T.)

**Keywords:** lung cancer, malignant pleural mesothelioma, traditional Chinese medicine, narciclasine, spheroids

## Abstract

Thoracic tumors, including lung adenocarcinoma (LUAD) and malignant pleural mesothelioma (MM), remain a leading cause of cancer-related deaths, primarily due to the challenges in treating advanced thoracic tumors. New drugs with optimal efficacy and minimal side effects are required. Traditional Chinese Medicine (TCM) compounds offer a promising alternative. Our study explores the effectiveness of the TCM compound Narciclasine against LUAD and MM. Spheroids derived from LUAD (A549, LuCa1) and MM (MSTO-211H, H2052/484) cell lines were treated with Narciclasine, and efficacy was evaluated through cell morphology analysis, intracellular adenosine triphosphate (ATPi) and lactate dehydrogenase (LDH) assays, RT-qPCR, and western blot analysis. Narciclasine reduced cell viability, with an 80% reduction in viability in MM. It induced cell apoptosis and inhibited proliferation. The IC_50_ values for Narciclasine ranged from 50 to 150 nM. In silico analysis identified shared gene targets between Narciclasine, LUAD, and MM. Narciclasine modulated the expression of genes associated with ferroptosis and cuproptosis, and the epithelial-to-mesenchymal transition. Narciclasine shows promising efficacy against LUAD and MM spheroid models. These findings warrant further investigation into the mechanisms of action and potential clinical application of Narciclasine for treating thoracic malignancies, offering hope for improved therapeutic options and patient outcomes.

## 1. Introduction

Thoracic malignancies, such as lung cancer and malignant pleural mesothelioma (MM), are highly aggressive cancers. In 2022, global incidence and mortality for lung cancer were approximately 2.5 million and 1.8 million, respectively. For MM, these figures were around 31,000 and 25,000.

Lung cancer is the leading cause of cancer-related deaths worldwide [1]. It is traditionally categorized into non-small cell lung cancer (NSCLC) and small cell lung cancer (SCLC), accounting for approximately 80% and 20% of cases, respectively [2]. The primary subtypes of NSCLC include adenocarcinoma (40% of cases), squamous cell carcinoma (25–30%), and large cell carcinoma (10–15%) [3]. Despite continuous advancements in treatment methods, the five-year survival rate for lung cancer remains below 20%, and only about 6% for advanced NSCLC [4]. Standard treatments include chemotherapy, radiotherapy, and targeted therapy. In recent years, immunotherapy has emerged as a promising treatment strategy, particularly for patients unresponsive to chemotherapy or targeted therapy. However, significant variations in treatment outcomes persist, with some patients achieving long-term disease control or remission and others showing limited benefit [5]. This discrepancy underscores the urgent need for identification or development of new effective treatments or combinations of existing ones.

Malignant pleural mesothelioma is a highly aggressive and lethal form of cancer that originates in the outer serosa of the lungs (pleura). It is predominantly associated with occupational exposure to asbestos [6]. MM is usually diagnosed at an advanced stage, with a five-year survival rate of 5–10% [7]. MM is classified into three main subtypes: epithelioid (75% of all cases), biphasic (15%), and sarcomatoid (10%), each exhibiting distinct epithelial–mesenchymal transition (EMT) phenotypes and varying resistance to treatment [8,9]. Patients with sarcomatoid and biphasic tumors have significantly worse overall survival compared to those with epithelioid tumors. Current treatment options for MM are limited, and the prognosis remains poor. Only relatively early-stage MM is considered for surgical treatment, but surgical resection is rarely complete and often requires adjuvant therapy. Most MM patients receive chemotherapy and other systemic therapies as their primary treatment [10]. While these modalities can alleviate symptoms and improve quality of life, median survival remains approximately 9–18 months [6]. Given the limited efficacy of current treatments and the poor prognosis for MM patients, there is an urgent need to identify more effective treatment options.

Traditional Chinese Medicine (TCM) is a branch of traditional medicine. While scientific research and evidence for the effectiveness of TCM are still limited, there has been a growing interest in investigating its safety and effectiveness, as well as the underlying mechanisms of action. Several studies suggest that TCM compounds may have potential anti-cancer properties, which could offer promising new treatment options for lung cancer and MM [11,12,13]. TCM herbs and compounds can inhibit tumor growth and metastasis, induce apoptosis, and enhance the sensitivity of cancer cells to chemotherapy and radiotherapy [14]. For example, a longitudinal study of over 100,000 newly diagnosed lung cancer patients found that TCM treatment was associated with a 32% reduction in mortality risk and improved overall survival [15]. Another study showed that TCM may also benefit elderly patients with advanced lung cancer, who achieved long-term survival through TCM treatment [16]. However, few studies have explored the application of TCM in NSCLC and MM [17], highlighting the need for further research into new and effective TCM-based therapeutic strategies for these cancer types.

Narciclasine, also known as Lycoricidinol, is a natural isocarbostyril alkaloid found in the Narcissus plant Amaryllidaceae [18,19]. It has been reported to exhibit anti-tumor properties across various cancer cell lines, including gastric, breast, oral, and esophageal cancer cells [20,21,22,23]. Narciclasine exerts its cytotoxic effect by inhibiting protein synthesis via the eukaryotic translation initiation factor 4A (eIF4A), ultimately inducing apoptosis [24,25,26]. This distinguishes it from conventional chemotherapeutic agents. Additionally, its anti-proliferative, anti-migratory, and anti-angiogenic properties make it a robust candidate for cancer treatment.

Preclinical studies have confirmed the efficacy of Narciclasine in various cancer models and its potential in overcoming chemoresistance, a major challenge for current cancer therapies. Despite its potent anti-cancer activity, the clinical development of Narciclasine has been limited by its narrow therapeutic index. Only a few studies have evaluated its effect on lung cancer, with some suggesting it can reduce cell viability in A549 cells and enhance drug sensitivity [27,28,29]. A recent study by Lee et al. [30] showed that Narciclasine reduces NSCLC spheroid viability and enhances their sensitivity to cisplatin by upregulating NOXA expression and inhibiting MCL1 translation. However, no study to date has clearly demonstrated its anti-cancer effects in MM or elucidated its underlying mechanisms of action.

Therefore, the aim of this study is to investigate the therapeutic potential of Narciclasine in NSCLC and MM using 3D spheroid tumor models. This approach may provide a better understanding of the efficacy and mechanisms of action of this TCM compound and inform future development of novel treatment options for these diseases.

## 2. Results

### 2.1. Narciclasine Induces Morphological Changes in Lung Adenocarcinoma and Malignant Mesothelioma Spheroids

After 72 h of Narciclasine treatment, tumor spheroids were imaged and analyzed to assess morphological changes. LUAD cells formed dense, irregular spheroids (Figure 1A), while MM cells formed more regularly shaped spheroids (Figure 2A). We evaluated morphological changes across increasing concentrations of Narciclasine (12.5 nM to 800 nM). MM spheroids displayed clear morphological changes at lower concentrations: MSTO-211H spheroids responded from 25 nM and H2052/484 spheroids from 50–100 nM (Figure 2A). These changes included loosening and disintegration of tight cell clusters, accompanied by a reduction in spheroid size. In contrast, LUAD spheroids (A549 and LuCa1) showed only slight reductions in size (Figure 1A).

Spheroid size was quantified using the sum area parameter (sum) and normalized to untreated spheroids to account for variability in initial spheroid size. The effect of Narciclasine at 440 nM was independent of initial spheroid size (Appendix A), indicating that the accessibility of the molecule was similar across conditions. LUAD spheroids showed only minimal changes in sum (Figure 1B), while H2052/484 (MM) spheroids showed a dose-dependent decrease (Figure 2B). These data suggest that Narciclasine induces partial disintegration of LUAD and MM spheroids, disrupting their compact form and promoting dispersion of dead cells around the spheroids, with a greater effect on MM spheroids.

### 2.2. Narciclasine Affects the Viability of Lung Adenocarcinoma and Pleural Mesothelioma Cells

In order to establish whether the morphological changes observed in the spheroids are linked to changes in cell viability, the cell viability and metabolism were assessed using LDH and ATPLite assays after 72 h of Narciclasine treatment. Narciclasine reduced cell viability in MM spheroids in a dose-dependent manner (Figure 2C; IC_50_: 81 nM for MSTO-211H, 166 nM for H2052/484) and ATP metabolism (Figure 2D; IC_50_: 46 nM for MSTO-211H, 83 nM for H2052/484), with MSTO-211H spheroids showing slightly greater sensitivity. In LUAD spheroids, A549 spheroids also showed reduced cell viability (Figure 1C; IC_50_: 145 nM), and ATP metabolism (Figure 1D; IC_50_: 121 nM), while LuCa1 spheroids showed reduced ATP metabolism (Figure 1D; IC_50_: 91 nM) LuCa1 cells showed LDH activity in the control condition (DMSO 0.2%) as high as that in the positive control obtained after treatment with Triton X-100 (Appendix A). This test could therefore not be used to estimate the cytotoxic effect of Narciclasine on LuCa1 cells. Interestingly, LuCa1 cells carry mutations not found in the A549 cell line, including in AURKA, a gene known to play a central role in kinetochore/chromatin-associated microtubule assembly in human cells [31]. The Aurora-A kinase was also reported to phosphorylate a subunit of LDH, increasing its activity in reducing pyruvate to lactate [32]. These results suggest that Narciclasine affects metabolism in both LUAD and MM spheroids, with greater efficacy observed in MM spheroids, and exerts cytotoxic effects in MM and A549 spheroids.

### 2.3. Narciclasine Affects Cell Proliferation and Cell Death in Lung Adenocarcinoma and Pleural Mesothelioma Spheroids

To further explore the effects of Narciclasine on tumor cells, we assessed cell proliferation and apoptosis at 50 nM Narciclasine. This dose is effective across all cell lines (IC_50_ for MSTO-211H: 46 nM, Figure 2D) while preserving cell viability after 72 h of treatment, and aligns with the National Cancer Institute reported average IC_50_ of 47 nM across a panel of 60 cancer cell lines [27].

Untreated MM spheroids exhibited significantly higher proliferation compared to LUAD spheroids (Figure 3A,B; Appendix A), with 8.8% ± 4.8 Ki67-positive cells for MM vs. 2.2% ± 1.4 for LUAD (*p* < 0.01). LuCa1 spheroids showed a non-significant decrease in Ki67-positive cells (2.35% ± 1.12 in controls vs. 1.07% ± 0.40 in treated LuCa1 spheroids, *p* = 0.25). In contrast, no consistent change was observed in treated MM spheroids (Figure 3B), indicating a limited anti-proliferative effect.

TUNEL staining revealed no significant increase in apoptosis in A549 and H2052/484 spheroids (Figure 3C,D; Appendix A). Increased apoptosis was observed in MSTO-211H (3.90% ± 2.03 for untreated MSTO-211H vs. 11.67% ± 9.72 for treated MSTO-211H, *p* = 0.0156), and LuCa1 spheroids (2.74% ± 2.08 in untreated LuCa1 vs. 4.88% ± 3.81 for treated LuCa1, *p* = 0.0156). These findings indicate that Narciclasine promotes apoptosis in MSTO-211H and LuCa1 cells, but not in all the evaluated cell lines, highlighting the need for larger data sets.

### 2.4. Identification of Narciclasine Gene Targets

To explore key genes potentially involved in the mechanism of action of Narciclasine, we conducted a network pharmacology analysis. We identified 308 potential gene targets linked to lung cancer and 62 to MM (Figure 4), with 61 targets common to both.

Gene set enrichment analysis (GSEA) was performed using EnrichR (https://maayanlab.cloud/Enrichr/; accessed on 7 February 2024) to analyze pathway enrichment and identify key regulatory elements associated with our gene set, based on the KEGG 2021 Human and MSigDB Hallmark 2020 datasets (Table 1).

The KEGG pathway revealed significant enrichment in pathways related to inflammation, survival signaling, and apoptosis, including the TNF signaling pathway, IL-17 signaling pathway, NF-kappa B pathway, and apoptosis signaling (Table 1). The MSigDB Hallmark gene set analysis further supported these findings, highlighting inflammatory response, p53 signaling, and EMT as key processes associated with Narciclasine treatment (Table 1). The enrichment of the inflammatory response gene set aligns with the KEGG results showing activation of TNF and IL-17 signaling, both of which are critical regulators of tumor-associated inflammation. Additionally, the enrichment of the p53 pathway suggests a role in cell cycle regulation and apoptosis, further supported by KEGG’s data.

### 2.5. Narciclasine Effects on the Ferroptosis and Cuproptosis Pathways

Several lines of evidence suggest functional and regulatory relationships between the TNF, IL-17, NF-κB, and p53 pathways and the mechanisms of ferroptosis and cuproptosis. Ferroptosis is an iron-dependent form of regulated cell death that involves the accumulation of reactive oxygen species (ROS) and lipid peroxidation. In a previous study [33], we showed that impaired iron-sulfur protein biogenesis disrupts mitochondrial respiration, DNA metabolism, and the proliferation of lung cancer and mesothelioma cells. Similarly, copper acts as a transition metal involved in redox reactions, contributing to ROS generation. Since Narciclasine has been reported to increase ROS production in A549 cells [30], we investigated whether it could affect the ferroptosis and cuproptosis pathways.

To explore this, we assessed the expression of 19 markers associated with ferroptosis and cupropotisis pathways in the four cell lines (MSTO-211H, H2052/484, A549, and LuCa1) treated or not for 72 h with 50 nM of Narciclasine. In both MM cell lines, most markers were upregulated after Narciclasine treatment (Figure 5, Appendix A), with the most notable fold increases observed for LIPT1 (3.8-fold in MSTO-211H, 4.5-fold in H2052/484), FDX1 (2.9-fold in MSTO-211H, 2.2-fold in H2052/484), and SLC7A5 (2.5-fold in MSTO-211H, and 2.6-fold in H2052/484). In contrast, CAV-1 expression was downregulated in both MM cell lines, with a fold decrease of 0.44 in MSTO-211H and 0.47 in H2052/484.

Some markers were differentially regulated between the two MM cell lines. While GPX4 expression remained unchanged in H2052/484, it was reduced in MSTO-211H following treatment (0.43-fold). This difference was even more pronounced for RRM2, which was upregulated in MSTO-211H (3.15-fold), and strongly downregulated in H2052/484 (0.11-fold). In LUAD cell lines, most markers were upregulated after Narciclasine treatment, with a more pronounced response in LuCa1 compared to A549 (Figure 5, Appendix A), particularly for SLC7A5. RRM2 was differently regulated by Narciclasine treatment in LUAD cells, being downregulated in LuCa1 (0.40-fold) and unchanged in A549 (1.04-fold).

To confirm transcriptional changes at the protein level, we analyzed the expression of three markers by western blot: SLC7A5, a ferroptosis marker upregulated in all four cell lines; RRM2, a ferroptosis marker differentially regulated across MM and LUAD cell lines; and DLAT, a cuproptosis marker upregulated in all four cell lines. As shown in Figure 6A, both MM cell lines expressed undetectable or very low levels of SLC7A5 protein. Basal SLC7A5 levels were higher in LUAD cells, especially in A549. Narciclasine treatment, however, did not significantly affect SLC7A5 protein levels. This discrepancy between mRNA and protein levels for SLC7A5 may reflect translational inhibition of mRNA. RRM2 protein levels mirrored transcriptional-level changes: increased in MSTO-211H cells, decreased in the H2052/484 and LuCa1 cells, and unchanged in A549 cells (Figure 6B). DLAT protein expression was slightly reduced in three of the four cell lines despite upregulated mRNA (Figure 6C), suggesting possible inhibition in translation of mRNA and/or enhanced proteosomal degradation.

### 2.6. Narciclasine Affects the EMT State of Cancer Cells

Based on the observed effects on spheroid morphology, viability, and results from the pathway enrichment analysis, we further investigated whether Narciclasine could alter the EMT state of tumor cells, a process crucial for tumor invasion and metastasis. We assessed the expression of eight EMT markers: E-cadherin (CDH1), N-cadherin (CDH2), vimentin (VIM), and the transcription factors SNAIL1, SNAIL2, ZEB1, ZEB2, and TWIST. In MSTO-211H spheroids, SNAIL1, SNAIL2, and ZEB2 were upregulated with Narciclasine treatment compared to untreated controls (Figure 7 and Appendix A). Little or no change was observed in CDH2, ZEB2, TWIST, and VIM. In H2052/484 spheroids, VIM, SNAIL1, and TWIST were downregulated, while ZEB2 and SNAIL2 increased slightly, and CDH2 or ZEB1 were unchanged. Notably, E-cadherin (CDH1) expression was undetectable in both MM cell lines.

In A549 and LuCa1 spheroids, Narciclasine treatment led to a decrease in CDH1 expression and an increase in CDH2, SNAIL1, and ZEB1 (Figure 7 and Appendix A). The expression of TWIST and VIM remained globally unchanged. Both SNAIL2 and ZEB2 were upregulated in treated A549, while ZEB2 was unchanged and SNAIL2 was undetectable in LuCa1.

## 3. Discussion

Thoracic tumors, including NSCLC and MM, remain the leading cause of cancer-related deaths worldwide. Despite therapeutic advances over the past decade, treating these cancers remains a significant challenge due to tumor heterogeneity (particularly for NSCLC), and increasing drug resistance. In MM, most patients are diagnosed when effective treatment options are limited. Currently, patients who cannot undergo surgery are primarily treated with symptom management. These limitations underscore the urgent need for more precise and effective therapies for thoracic tumors.

Narciclasine, a TCM compound, has been previously reported to have anti-inflammatory effects, reduce oxidative stress, and mitigate organ damage. In recent studies, it has been reported to exert anti-tumor activity by inhibiting tumor cell growth and inducing apoptosis in several cancer types, including triple-negative breast cancer, brain cancer, and primary lymphoma [19,25,26]. In gastric cancer, Narciclasine significantly improves survival rates by inducing autophagy-mediated apoptosis through the Akt/mTOR classic pathway [34]. Recently, Lee et al. [30] demonstrated that Narciclasine reduces NSCLC spheroid viability and increases their sensitivity to cisplatin by inducing apoptosis via upregulation of NOXA expression and inhibition of MCL1 translation. In our study, Narciclasine exhibits a similar inhibitory effect on the metabolism of both LUAD and MM spheroids, with up to 60–70% reduction. Narciclasine demonstrates a stronger anti-cancer effect on MM spheroids, with up to 80–85% reduction in viability, compared to A549 spheroids (50%). Interestingly, the reduction in MM spheroid viability appeared to be driven by increased apoptosis rather than decreased proliferation. Moreover, we found that Narciclasine also altered the expression of key markers associated with the ferroptosis and cuproptosis pathways in MM spheroids. Notably, dihydrolipoamide S-acetyltransferase (DLAT) levels were reduced after Narciclasine treatment in both MM cell lines. DLAT is a core enzyme of the mitochondrial pyruvate dehydrogenase complex that supports mitochondrial function as well as the Warburg effect in cancer cells [35]. Its suppression has been linked to inhibition of TCA cycle metabolism and glutathione synthesis in medulloblastoma [36] and impaired tumor progression in hepatocarcinoma [37]. Our findings support a hypothesis where downregulation of DLAT contributes to the cytotoxic effects of Narciclasine treatment in MM cells. This result needs to be confirmed by measuring the oligomerized form of DLAT. Indeed, excess intracellular copper in the cytoplasm binds to the lipoylated form of DLAT and leads to abnormal oligomerization of this protein that causes cell death [38]. Surprisingly, we observed that Narciclasine induces an increased expression of the Ribonucleotide reductase subunit M2 (RRM2) gene in MSTO-211H cells. This gene encodes the M2 catalytic subunit of the ribonucleoside-diphosphate reductase, an enzyme that catalyzes the formation of deoxyribonucleotides from ribonucleotides. RRM2, upregulated in different cancers, is a critical enzyme in cell proliferation [12,34,39], and has been shown to exert anti-ferroptotic activity in some cancer cells [40]. Its upregulation in MSTO-211H cells correlates with the increased proliferation rate observed following Narciclasine treatment (Figure 4B), suggesting a compensatory response that may explain the cytotoxic effect of Narciclasine in MM cells. In contrast, RRM2 protein levels were reduced following Narciclasine treatment in the LUAD LuCa1 cells, in line with reduced cell viability (as indicated by intracellular ATP levels) and decreased cell proliferation of these cells. In contrast to MM spheroids, no significant effects on cell apoptosis or expression of ferroptosis and cuproptosis markers were observed in A549 cells treated with 50 nM Narciclasine, in line with the absence of effect observed on the A549 viability at this concentration (Figure 1C). Lung adenocarcinoma cells often develop mechanisms of resistance to ferroptosis, especially with KRAS mutation [41,42]. A549 cells contain a specific mutation in the KRAS gene, the KRAS G12S mutation, that could partly explain the lower susceptibility to Narciclasine. Differences in mesothelioma and lung adenocarcinoma could also be partly explained by the mesenchymal phenotype of these cells. Tumor cells with a mesenchymal phenotype, such as mesothelioma cells MSTO-211H and H2052/484, may be particularly sensitive to ferroptosis cell death, while cells with an epithelial phenotype, such as LUAD A549 cells, are less sensitive.

Our results also suggest that Narciclasine may modify the EMT phenotype of MM and LUAD spheroids. Narciclasine treatment was associated with increased expression of SNAIL1, ZEB1/ZEB2, and CDH2, along with decreased expression of CDH1, indicating a shift towards a mesenchymal state. This phenotype is typically associated with enhanced migration and invasion capabilities, promoting metastasis, and resistance to apoptosis and therapeutic interventions. Our findings differ from those reported by Shieu et al. [22] in oral squamous cell carcinoma, where Narciclasine promoted expression of the epithelial marker ZO-1 and reduced expression of mesenchymal markers (CDH2, and β-catenin). Several factors may explain these differences: EMT is highly context-dependent, and variables like tumor type, model system (2D vs. 3D), and the endpoint measured (mRNA vs. protein) may influence how cells respond to Narciclasine. Previous unpublished data from our group suggest that EMT marker expression differs between monolayer and in vivo models, with 3D spheroids providing a more physiologically relevant model of tumor behaviour. In our study, we analysed mRNA expression, while Shieu et al. [22] measured protein levels. Since mRNA and protein levels do not always correlate, protein-level validation will be necessary to confirm our findings. Overall, these differences highlight the complexity of EMT regulation and the importance of considering tumor type, microenvironment, and methodology when interpreting the effects of Narciclasine. Further investigation is therefore necessary to clarify its precise role in EMT and across different cancer models.

Through network pharmacology, an effective tool for comprehensive analysis of the complex interactions between therapeutic targets and potential drugs, we identified 61 potential target genes common to Narciclasine in both lung cancer and MM. Most of the genes are enriched in the TNF, NF-kappa B, and IL-17 signaling pathways, all of which are closely related to tumor apoptosis and proliferation. This integrated analysis highlights that Narciclasine may exert its effects by disrupting inflammatory signaling, promoting apoptosis, and modulating EMT, which could contribute to its anti-cancer activity in lung cancer and mesothelioma. Further studies are needed to identify the signalling pathways targeted by Narciclasine, and above all to evaluate the combined effect of Narciclasine with chemotherapeutic agents (such as cisplatin) on thoracic tumors, mesothelioma, and lung cancer. This combination could improve the efficacy of treatments used at lower doses, thereby limiting side effects and the development of resistance mechanisms. Although numerous pre-clinical studies have shown very encouraging anti-cancer activity of Narciclasine, no clinical trials appear to have been initiated to date. Its limited availability is certainly one reason for this. However, several recent studies have described the chemical synthesis of Narciclasine and its analogues [19,43,44], paving the way for clinical studies.

## 4. Materials and Methods

### 4.1. Cell and Spheroid Culture

Human cell lines A549 and MSTO-211H were purchased from the American Type Culture Collection (Manassas, VA, USA). LuCa1 is a cancer cell line established and characterized in the laboratory [45] from a 79-year-old woman diagnosed with untreated stage IV LUAD. H2052/484 cells were derived in the laboratory from H2052 cells [46]. Briefly, 1 × 10^6^ H2052 cells were implanted in the thoracic cavity of Nude-Foxn1nu nu/nu athymic mice. The orthotopic tumor was explanted and mechanically dissociated 102 days later to produce the H2052/484 cell line. All cell lines were cultured in Roswell Park Memorial Institute (RPMI) 1640 (Life Technologies, Carlsbad, CA, USA) supplemented with 10% (*v*/*v*) heat-inactivated fetal bovine serum (FBS), 10 mM HEPES, and 100 U/mL-100 µg/mL Penicillin-Streptomycin (Sigma-Aldrich Chemie GmbH, Buchs, Switzerland) (complete RPMI). To produce spheroids, after trypsinization, MSTO-211H and H2052/484 cells were seeded in Akura™ 96- or 384-well Spheroid Microplates (InSphero, Schlieren, Switzerland) at 1500 cells per well. LuCa1 and A549 cells were seeded at 3000 cells per well. All cells were cultured in complete RPMI at 37 °C in 5% CO_2_.

### 4.2. Narciclasine Treatment

Narciclasine (1 mg; MedChemExpress, Monmouth, NJ, USA) was dissolved in sterile dimethyl sulfoxide (DMSO) (Sigma-Aldrich Chemie GmbH, Buchs, Switzerland) to generate a 1 mM stock solution. Working concentrations of Narciclasine (400 nM, 200 nM, 100 nM, 50 nM, 25 nM and 12.5 nM) were prepared by diluting the stock solution in complete RPMI medium with 1% heat-inactivated FBS. After 3 days of culture, A549, LuCa1, MSTO-211H, and H2052/484 spheroids (3000 and 1500 cells/spheroid) were incubated with Narciclasine at the indicated concentrations for 72 h. Negative control spheroids were incubated for 72 h in RPMI + 1% FBS medium containing 0.08% DMSO (corresponding to the final DMSO concentration in all Narciclasine treatments). Positive cytotoxic control spheroids were incubated for 72 h in RPMI + 1% FBS medium containing 0.1% Triton X-100 (AppliChem, Darmstadt, Germany).

### 4.3. Cell Viability and Cytotoxicity Assays

Cell viability was assessed by measuring intracellular adenosine triphosphate (ATPi) levels using the luminescence-based ATPliteTM assay (PerkinElmer AG, Schwerzenbach, Switzerland) as described by Gendre et al. [45]. Cytotoxicity was measured using the colorimetric LDH assay (LDH Assay Kit, ab102526) (Abcam, Cambridge, UK) according to manufacturer instructions. After 72 h of treatment, 10 µL of supernatant was collected from each well of the Akura™ 96- or 384-well Spheroid Microplates and incubated with 10 μL of Reaction Mixture. Absorbance in each well was recorded in kinetic mode (36 cycles of measurements) at 440 nm using a microplate reader (Fluostar Optima, BMG Labtech, Ortenberg, Germany) on the READS platform. Considering untreated cells as the negative control and Triton 100-X as the positive control, relative LDH production was calculated using the following equation: Relative LDH = (X − Negative control)/(Positive control − Negative control)
where X is the absorbance value of the sample at 440 nm.

### 4.4. Quantification of Cell Proliferation and Apoptosis

Cell proliferation was evaluated by immunolabeling cell nuclei with an anti-Ki67 antibody (Cell Signaling Technology, Danvers, MA, USA), as described by Mueggler et al. [47]. The proportion of proliferative cells was determined by calculating the ratio of Ki67-positive cells to the total number of Hoechst-stained cells.

Cell apoptosis was analyzed using the TUNEL assay (In Situ Cell Death Detection Kit, TMR red, Sigma-Aldrich Chemie GMBH) following the manufacturer’s instructions. Imaging was performed using a Zeiss Apotome (Jena, Germany), at 20× magnification (Axiovision 4.6). The proportion of apoptotic cells was calculated as the percentage of TUNEL and Hoechst-stained cells.

### 4.5. RNA Extraction and Quantitative RT-PCR (qPCR)

For each condition, a total of 512 spheroids cultured in two agarose casts were recovered after centrifugation of the inverted agarose casts at 100x *g* for 5 min. Total RNA was extracted using the InviTrap Spin Universal RNA Mini Kit (Invitek Molecular GmbH, Berlin, Germany), following the manufacturer’s instructions. cDNA was synthesized from 0.5 µg of total RNA using reverse transcriptase (Invitrogen, Life Technologies, Carlsbad, CA, USA). Quantitative PCR from 50 ng cDNA was performed in triplicate using Takyon™ No Rox SYBR Master Mix dTTP Blue (Eurogentec, Seraing, Belgium) and indicated primers (Table 2).

Gene expression was quantified using the double delta Ct (ΔΔCt) method and normalized to housekeeping genes.

### 4.6. Network Pharmacology Analysis

The active ingredient of Narciclasine was identified using the TCM Systems Pharmacology Database (TCMSP, https://tcmsp-e.com/tcmsp.php, accessed on 7 February 2024). Target genes were retrieved from three databases: GeneCards (https://www.genecards.org/), Online Mendelian Inheritance in Man (OMIM, https://omim.org/), and CTD Database Commons (CTD, https://ngdc.cncb.ac.cn/databasecommons/database/id/754, accessed on 7 February 2024). Genetic information was processed and analysed using Cytoscape 3.8.2 software, R software 4.1.0, Venny 2.1.0 (https://bioinfogp.cnb.csic.es/tools/venny/index.html, accessed on 7 February 2024).

### 4.7. Statistical Analysis

All statistical analyses were performed using GraphPad Prism (version 8.0). Data are presented as mean ± standard deviation (SD), unless stated otherwise. The Mann–Whitney test was used to compare two groups. Differences were considered statistically significant at *p* < 0.05.

## 5. Conclusions

In summary, Narciclasine, a TCM compound, shows promise against lung cancer and MM, even when used as a single agent. Our study, using 3D spheroid cell culture technology, supports its potential as a therapeutic agent. Most of the effects of Narciclasine on LUAD and MM spheroids were evaluated using a concentration of 50 nM over a period of 3 days, known not to cause toxicity in non-malignant cells [27]. This low concentration close to the IC_50_ measured on MM spheroids in the first part of our study was chosen in order to identify the effects and signalling pathways impacted by this concentration of Narciclasine with a view to using Narciclasine in combination with other compounds (e.g., chemotherapeutic agents such as platinum compounds). The moderate sensitivity of spheroids to Narciclasine alone can be explained by the use of this low concentration. However, it is interesting to note that this low concentration initiates significant changes in the ferroptosis, cuproptosis and EMT signalling pathways, thus suggesting that Narciclasine at this concentration could be combined with other anti-cancer therapies. To further validate these findings, we are currently developing a more complex 3D spheroid model that incorporates both tumor cells and cancer-associated fibroblasts (CAFs), thereby integrating the stromal component of the tumor microenvironment. This approach will enable us to better assess the impact of Narciclasine in a more physiologically relevant setting. We also plan to evaluate Narciclasine’s toxicity on non-cancerous lung cells within a similar 3D spheroid model to determine its selectivity and potential effects on healthy tissue. Additionally, as Lee et al. [30] showed a synergetic effect of Narciclasine with cisplatin, and as platinum-based chemotherapy is commonly used to treat mesothelioma, we plan to investigate the combinational anticancer efficacy of cisplatin and Narciclasine in MM. Our findings warrant further investigation into the mechanism of action and potential clinical application of Narciclasine in treating thoracic malignancies, offering hope for improved treatment options and improved patient outcomes.

## Figures and Tables

**Figure 1 ijms-26-10127-f001:**
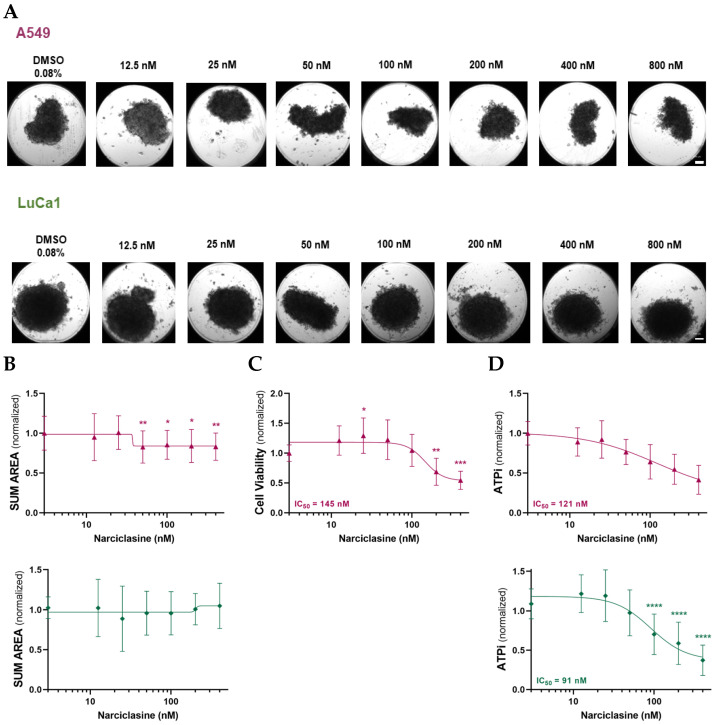
Narciclasine impairs the morphology and the viability of lung adenocarcinoma (LUAD) spheroids. Effects of increasing concentrations of Narciclasine on A549 and LuCa1 LUAD spheroids treated for 72 h on (**A**) morphology (representative phase contrast images of spheroids. Scale bar: 100 µm), (**B**) Spheroid size quantified by normalized Sum area, (**C**) Cell viability assessed using the LDH assay (**D**) Cell metabolism measured via ATPlite assay. Data are presented as mean ± standard deviation of three biological replicates. Two-tailed unpaired *t*-test statistical analysis was performed, comparing each concentration to controls. * *p* < 0.05; ** *p* < 0.01; *** *p* < 0.001; **** *p* < 0.0001.

**Figure 2 ijms-26-10127-f002:**
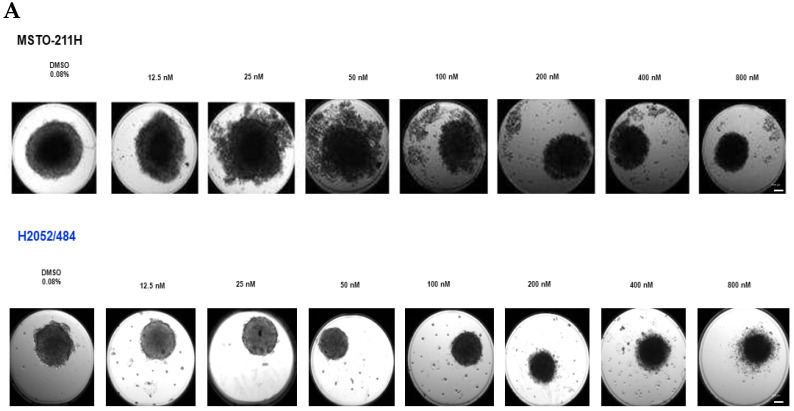
Narciclasine impairs the morphology and viability of mesothelioma (MM) spheroids. Effects of increasing concentrations of Narciclasine on MSTO-211H and H2052/484 MM spheroids treated for 72 h on (**A**) morphology (representative phase-contrast images of spheroids. Scale bar: 100 µm), (**B**) Spheroid size quantified by normalized sum area, (**C**) Cell viability assessed using LDH assay. (**D**) Cell metabolism measured via ATPlite assay. Data are presented as mean ± standard deviation of three biological replicates. Two-tailed unpaired *t*-test statistical analysis was performed, comparing each concentration to controls. * *p* < 0.05; ** *p* < 0.01; *** *p* < 0.001; **** *p* < 0.0001.

**Figure 3 ijms-26-10127-f003:**
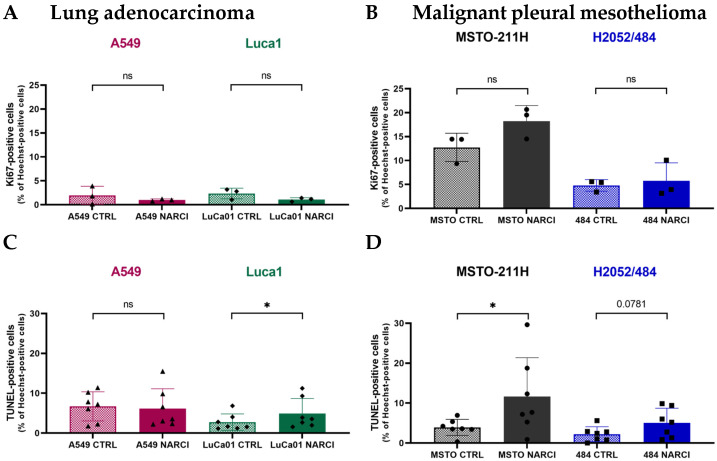
Effects of Narciclasine on cell proliferation and apoptosis in LUAD and MM spheroids. (**A**,**B**) Quantification of proliferating cells based on the percentage of Ki-67-positive cells over total Hoechst-stained nuclei in LUAD (**A**) and MM (**B**) spheroids treated with 50 nM Narciclasine for 72 h. (**C**,**D**) Quantification of apoptotic cells based on percentage of TUNEL-positive cells over total Hoechst-stained nuclei in LUAD (**C**) and MM (**D**) spheroids after 72-h treatment with 50 nM Narciclasine. Data are normalized to untreated control (CTRL) spheroids and presented as mean ± standard deviation of three biological replicates. Two-tailed paired *t*-test statistical analysis was performed, comparing the effect on cells in the presence (NARCI) and absence (CTRL) of Narciclasine. ns—*p* ≥ 0.05; * *p* < 0.05.

**Figure 4 ijms-26-10127-f004:**
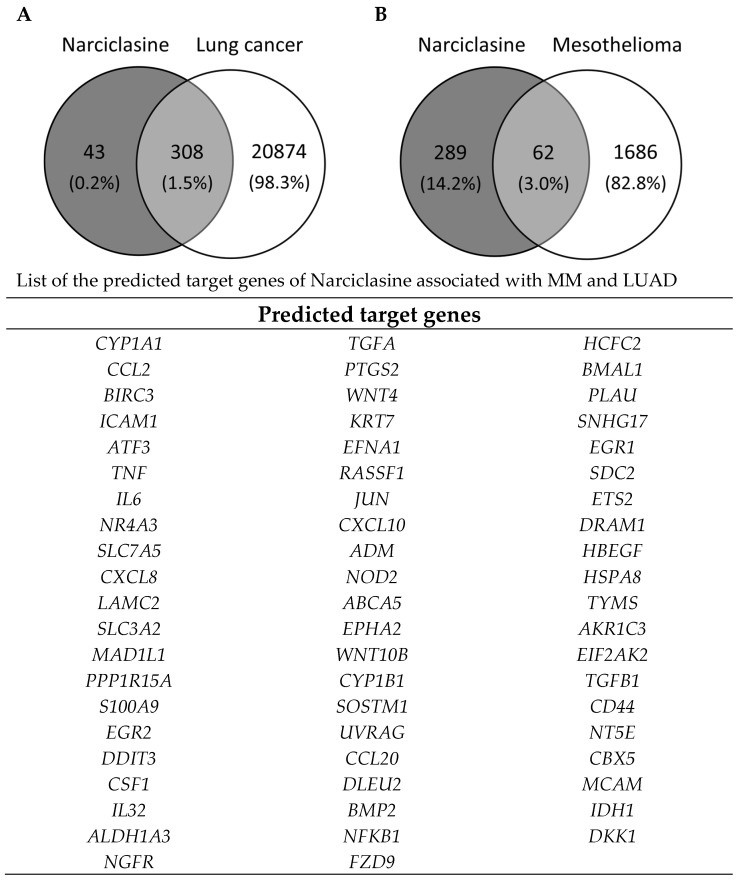
Bioinformatics identification of Narciclasine gene targets. Determination of the number of common targets between Narciclasine and lung cancer (**A**), and Narciclasine and MM (**B**). List of the predicted target genes of Narciclasine associated with both MM and LUAD.

**Figure 5 ijms-26-10127-f005:**
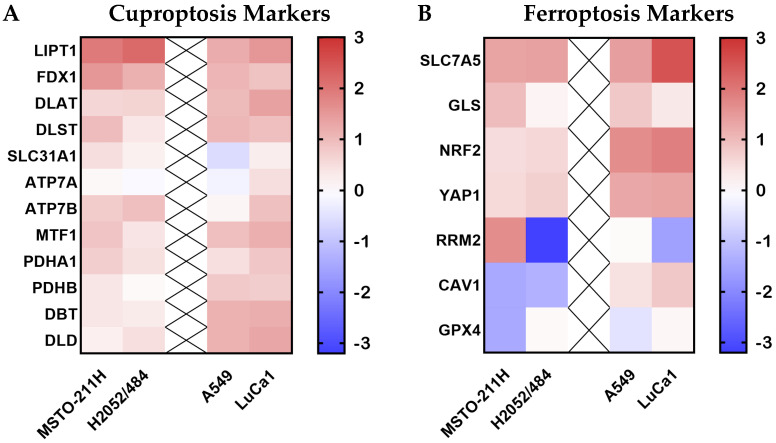
Effects of Narciclasine on the mRNA expression of cuproptosis and ferroptosis markers. (**A**) Heatmaps showing Log2 fold changes in mRNA expression of markers involved in cuproptosis (**A**) and ferroptosis (**B**) pathways following 72-h treatment with 50 nM Narciclasine of MM and LUAD cells.

**Figure 6 ijms-26-10127-f006:**
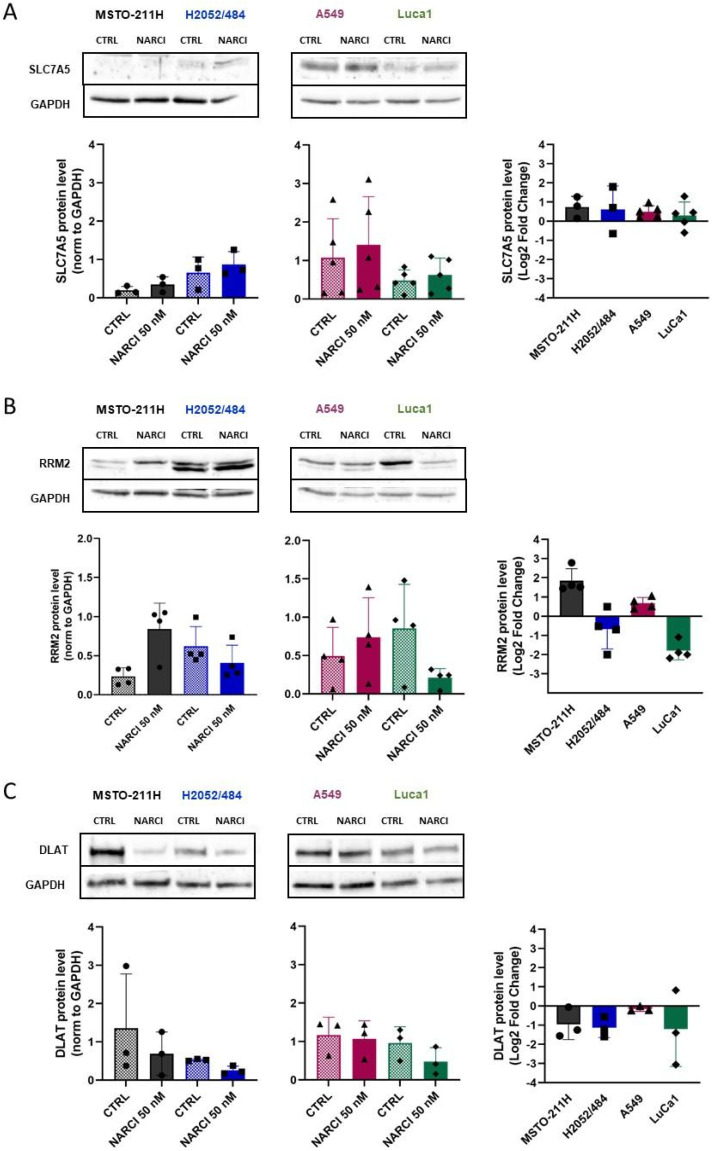
Effects of Narciclasine on the protein expression of cuproptosis and ferroptosis markers. Western blot analysis of ferroptosis markers (**A**) SLC7A5 and (**B**) RRM2, and cuproptosis marker (**C**) DLAT in control and Narciclasine-treated MM and LUAD cells following 72 h of treatment with 50 nM Narciclasine. For each marker, the relative amount of protein for the control and treated conditions is indicated, as well as the variation in expression level between the treated and control conditions (right panels) presented on a Log2 scale. Data are presented as mean ± SD from 3–4 independent experiments.

**Figure 7 ijms-26-10127-f007:**
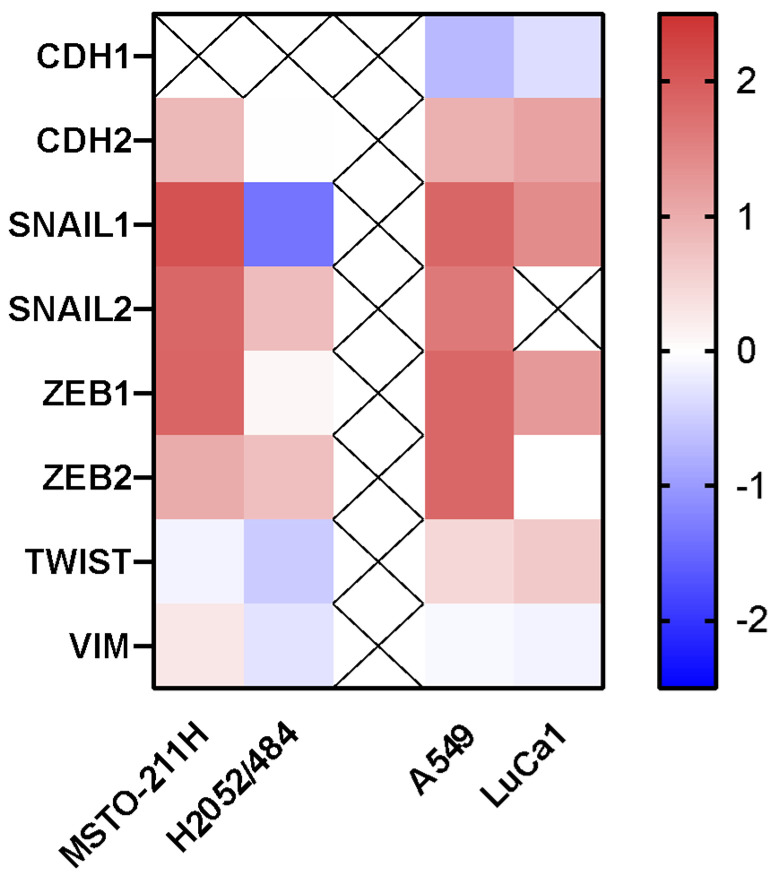
Effects of Narciclasine on the expression of epithelial-to-mesenchymal (EMT) markers. Heatmap showing Log2 fold changes in mRNA expression of EMT markers in MM and LUAD cells following 72 h of treatment with 50 nM Narciclasine.

**Table 1 ijms-26-10127-t001:** Enriched KEGG pathway, and Hallmark genes obtained from KEGG 2021 Human and MSigDB Hallmark 2020 databases.

**KEGG Pathway**			
**Entry**	**Pathway**	**Adjusted *p*-Value**	**Genes**
hsa04668	TNF signaling pathway	1.48 × 10^−13^	*CXCL10*; *JUN*; *IL6*; *CSF1*; *CCL20*; *CCL2*; *NOD2*; *PTGS2*; *TNF*; *NFKB1*; *ICAM1*; *BIRC3*
hsa04657	IL-17 signaling pathway	2.36 × 10^−11^	*CXCL10*; *JUN*; *IL6*; *CXCL8*; *CCL20*; *CCL2*; *PTGS2*; *TNF*; *S100A9*; *NFKB1*
hsa04064	NF-kappa B signaling pathway	4.19 × 10^−7^	*CXCL8*; *PLAU*; *PTGS2*; *TNF*; *NFKB1*; *ICAM1*; *BIRC3*
hsa04210	Apoptosis	2.75 × 10^−5^	*NGFR*; *JUN*; *DDIT3*; *TNF*; *NFKB1*; *BIRC3*
**MSigDB Hallmark**			
	**Hallmark Name**	**Adjusted *p*-Value**	**Genes**
	Inflammatory Response	1.95 × 10^−11^	*CXCL10*; *IL6*; *CXCL8*; *CSF1*; *CCL20*; *CCL2*; *EIF2AK2*; *ADM*; *NOD2*; *NFKB1*; *ICAM1*; *HBEGF*
	p53 Pathway	7.10 × 10^−9^	*PPP1R15A*; *JUN*; *BMP2*; *DRAM1*; *DDIT3*; *TGFA*; *SLC3A2*; *ATF3*; *EPHA2*; *HBEGF*
	Epithelial–Mesenchymal Transition	5.97 × 10^−8^	*IL32*; *NT5E*; *JUN*; *IL6*; *CXCL8*; *LAMC2*; *TGFBI*; *DKK1*; *CD44*

**Table 2 ijms-26-10127-t002:** Sequences of the primers used in quantitative PCR.

Gene	Forward Primer	Reverse Primer
*ATP7A*	TGACCCTAAACTACAGACTCCAA	CGCCGTAACAGTCAGAAACAA
*ATP7B*	GGCCGTCATCACTTATCAGCC	GGGAGCCACTTTGCTCTTGA
*Cav1*	CATCCCGATGGCACTCATCTG	TGCACTGAATCTCAATCA
*CDH1*	CCCGGGACAACGTTTATTAC	GCTGGCTCAAGTCAAAGTCC
*CDH2*	GGTGGAGGAGAAGAAGACCAG	GGCATCAGGCTCCACAGT
*DBT*	CAGTTCGCCGTCTGGCAAT	CCTGTGAATACCGGAGGTTTTG
*DLD*	CTCATGGCCTACAGGGACTTT	GCATGTTCCACCAAGTGTTTCAT
*DLAT*	ACTCCCCAGCCTTTAGCTC	CAATCCCTTTCTCTACTGCCAAC
*DLST*	GAACTGCCCTCTAGGGAGAC	AACCTTCCTGCTGTTAGGGTA
*EEF1A1*	ACTACCCCTAAAAGCCAAAATGG	GGTGGACTTGCCCGAATCTA
*FDX1*	CCACTTTATAAACCGTGATGGTG	ACATGCACCAAAGCCATCAA
*GAPDH*	GCACAAGAGGAAGAGAGAGACC	AGGGGAGATTCAGTGTGGTG
*GLS*	TCTACAGGATTGCGAACGTCT	CTTTGTCTAGCATGACACCATCT
*GPX4*	GAGGCAAGACCGAAGTAAACTAC	CCGAACTGGTTACACGGGAA
*GUSB*	ACGTGGTTGGAGAGCTCATT	CTCTGCCGAGTGAAGATCCC
*HPRT1*	ACAGGACTGAACGTCTTGCTCG	TGATGTAATCCAGCAGGTCAGCA
*LIPT1*	CCTCTGTTGTAATTGGTAGGCAT	CTGGGGTTGGACAGCATTCAG
*MTF1*	CACAGTCCAGACAACAACATCA	GCACCAGTCCGTTTTTATCCAC
*NRF2*	AGGTTGCCCACATTCCCAAA	ACGTAGCCGAAGAAACCTCA
*PDHA1*	TGGTAGCATCCCGTAATTTTGC	ATTCGGCGTACAGTCTGCATC
*PDHB*	AAGAGGCGCTTTCACTGGAC	ACTAACCTTGTATGCCCCATCA
*RPLP0*	GAAGACAGGGCGACCTGGAAG	GCGCATCATGGTGTTCTTGCC
*RRM2*	CTGGCTCAAGAAACGAGGACT	ACATCAGGCAAGCAAAATCACA
*SLC7A5*	CCGTGCCGTCCCTCG	AGAAGGCGTAGAGCAGCGTC
*SLC31A1*	GGGGATGAGCTATATGGACTCC	TCACCAAACCGGAAAACAGTAG
*SNAIL-1*	GCTGCAGGACTCTAATCCAGA	ATCTCCGGAGGTGGGAT
*SNAIL-2*	TGGTTGCTTCAAGGACACAT	GTTGCAGTGAGGGCAAGAA
*TBP*	GCCCGAAACGCCGAATATA	CGTGGCTCTCTTATCCTCATGA
*TWIST*	CGGCCAGGTACATCGACT	CATCTTGGAGTCCAGCTCGT
*VIM*	AGATGGCCCTTGACATTGAG	TGGAAGAGGCAGAGAAATCC
*YAP1*	AGAGGCTGCGGCTGAAAC	TTGCTGTGCTGGGATTGATA
*ZEB-1*	GCCAACAGACCAGACAGTGTT	CAGGAAAGGAAGGGCAAGA
*ZEB-2*	CAAGAGGCGCAAACAAG	AACCTGTGTCCACTACATTGTCA

## Data Availability

The original contributions presented in this study are included in the article/Appendix A. Further inquiries can be directed to the corresponding author.

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
