# Peer review of "Narciclasine as a Novel Treatment for Lung Cancer and Malignant Pleural Mesothelioma: Insights from 3D Tumor Spheroid Models"

_ijms, 2025, doi:10.3390/ijms262010127_

Round 1
Reviewer 1 Report
Comments and Suggestions for Authors
This study aims to evaluate the use of narciclasine as a therapeutic approach for lung and mesothelioma cancer. It focuses on further expanding the current studies available with this drug by providing a more mechanistical description of how narciclasine treatment affects molecular pathways using tumor spheroids that provide a more physiological in vitro model.
- It would be good to highlight in the introduction the advantages of the use of spheroids over a 2D culture as this is the increased value over previous studies.
- The description of results could be improved by better organizing the appearance of the mention of the results to correspond with the order of how they are being shown in the figures.
- It is appreciated that the results are provided in full detail in the supplementary materials. It is shown an expected biological behavior with the variability of the bars and lines, my only observation would be to consider replacing (repeating the assay) Figure 1, panel C corresponding to LuCa1 Viability, as that variability across the doses and the wide fold change appear to be related to experimental issues and inconsistencies.
- In lines 152-154 a mutation in AURKA gene is mentioned and it could be of interest to further develop this idea there or in the discussion as it is a proposed mechanism by the authors to explain the differences between cell lines.
- From results 2.3 all the doses of narciclasine are kept at 50nM for all the cell lines while this dose may appear to be low based on the previous results as LUAD cell lines doesn’t respond very well at this dose. It would be interesting to mention in the discussion this issue as this low dose can explain some of the results that showed low response to the drug.
- In lines 159-162 the dose of narciclasine is compared with a database of the National Cancer Institute. This is a very interesting and informative mention, but it needs to be taken with caution since a difference in response to a drug can vary between 2D and 3D culture models.
- Line 186 mentions apoptosis “in all patients” when it should be “the evaluated cell lines”.
- Results 2.4 and Figure 4 should be better explained as there is a lack of context to understand the importance of this analysis. For example, mentioning the origin of the data used and giving a brief description of the relevance of the finding of those genes. Additionally, Figure 4 needs the letters of the panels (A-C) and probably the table of genes can be modified to show the 50 most relevant genes for LUA and MM instead of leaving out completely LUA without a clear reason.
- In Results 2.5 regarding the WBs, it is mentioned the lack of correlation of these markers with the mRNA results. Have in mind that 72 hours can be already late to detect changes since that is the time you are observing a very low viability in your first results. The changes in mRNA/proteins are detectable before the death mechanisms are quantified and would be better to use shorter times of treatment for expression assays.
- It would be valuable to have in Discussion a mention of the cellular/tissue differences between the LUAD and MM cancer models since the response to narciclasine appears to be different between them.
Reviewer 2 Report
Comments and Suggestions for Authors
The Authors demonstrated that narciclasine reduced cell viability, with an 80% reduction in viability in MM. It induced cell apoptosis and inhibited proliferation. The IC50 values for Narciclasine ranged from 50 to 150 nM. In silico analysis identified shared gene targets between Narciclasine, LUAD, and MM.
- Statistical analysis should be provided for all quatified data in Fig. 1 and 2.
- Fig. 6C - DLAT oligomerization is a major marker of cuproptosis (e.g., doi: 10.1016/j.redox.2025.103552). The results should be discussed accordingly, or the Authors should consider to assess DLAT oligomers e.g., by confocal microscopy.
- Can the Authors provide any additional validation of the results shown in the Fig. 7?
- I would recommend to provide a potential model of activity based on obtained results.
Discussion of the results is sufficient, although more novel and up-to-date references in the field should be additionally considered by the Authors.
Reviewer 3 Report
Comments and Suggestions for Authors
- Although the anti-tumor effect of Narciclasine in gastric cancer, breast cancer and other tumors is mentioned, it does not systematically integrate the research gaps in thoracic tumors. For example, it fails to compare the research progress of other traditional Chinese medicine compounds (such as artemisinin derivatives and matrine) in lung cancer/MM, making it impossible to highlight the uniqueness of Narciclasine; at the same time, it only briefly mentions the limitation of its "narrow therapeutic index" and does not analyze the specific obstacles of this issue in clinical transformation (such as pharmacokinetic characteristics and toxic targets), which weakens the clinical orientation of the research.
- The overall design revolves around the sequence of "Narciclasine treatment → phenotypic detection → mechanism exploration", but each part lacks a logical closed loop. For example, in the mechanism research, pathways such as TNF and NF-κB are first screened through network pharmacology, and then markers of ferroptosis/ cuproptosis are detected, with no clear causal relationship established between the two.
- Cell viability assays used both LDH (cytotoxicity) and ATPlite (metabolic activity), but did not analyze the consistency of the results from the two methods (for example, in some cell lines, ATPlite showed decreased metabolism but LDH had no significant changes), nor did they rule out the interference of drugs on the detection reagents themselves (such as whether Narciclasine affects ATPase activity).
- It was found in the article that the inhibitory effect of Narciclasine on MM spheres (with an 80% reduction in viability) was significantly better than that on LUAD (a 50% reduction in A549), but the differences in the underlying molecular mechanisms were not analyzed.
- The format of IC50 in the abstract is not standard, and the same goes for that in the figures. It should be IC50.
- For the concentrations in Figure 1, it is recommended to use dots; the significance analysis is compared with that data. Figure 2 has the same problem as mentioned above.
- The tables in the article are numbered in the order of their appearance.
Round 2
Reviewer 2 Report
Comments and Suggestions for Authors
The comments have been addressed sufficiently.
Reviewer 3 Report
Comments and Suggestions for Authors
The quality of the article has improved after revision, and it is recommended for acceptance.